# Biochemical, genetic and transcriptional characterization of multibacteriocin production by the anti-pneumococcal dairy strain *Streptococcus infantarius* LP90

**Cristina Campanero[1], Estefanía Muñoz-Atienza[1¤], Dzung B. Diep[2], Javier Feito[1], Sara Arbulu[1], Rosa del Campo[3], Ingolf F. Nes[2], Pablo E. Hernández[1], Carmen Herranz[1], Luis M. Cintas[1]***

**1** Grupo de Seguridad y Calidad de los Alimentos por Bacterias Lácticas, Bacteriocinas y Probióticos (Grupo SEGABALBP), Sección Departamental de Nutrición y Ciencia de los Alimentos (Nutrición, Bromatología, Higiene y Seguridad Alimentaria), Facultad de Veterinaria, Universidad Complutense de Madrid, Madrid, Spain, **2** Department of Chemistry, Biotechnology and Food Science, Norwegian University of Life Sciences (NMBU), Ås, Norway, **3** Servicio de Microbiología, Hospital Universitario Ramón y Cajal & Instituto Ramón y Cajal de Investigación Sanitaria (IRYCIS), Madrid, Spain

¤ Current address: Centro de Investigación en Sanidad Animal (CISA), Instituto Nacional de Investigación y Tecnología Agraria y Alimentaria (INIA), Valdeolmos, Madrid, Spain
* lcintas@vet.ucm.es

**Data Availability Statement:** All relevant data are within the paper and its Supporting Information files.

## Abstract

*Streptococcus pneumoniae* infections are one of the major causes of morbidity and mortality worldwide. Although vaccination and antibiotherapy constitute fundamental and complementary strategies against pneumococcal infections, they present some limitations including the increase in non-vaccine serotypes and the emergence of multidrug-resistances, respectively. Ribosomally-synthesized antimicrobial peptides (*i.e.* bacteriocins) produced by Lactic Acid Bacteria (LAB) may represent an alternative or complementary strategy to antibiotics for the control of pneumococal infections. We tested the antimicrobial activity of 37 bacteriocinogenic LAB, isolated from food and other sources, against clinical *S. pneumoniae* strains. *Streptococcus infantarius* subsp. *infantarius* LP90, isolated from Venezuelan water-buffalo milk, was selected because of its broad and strong anti-pneumococcal spectrum. The *in vitro* safety assessment of *S. infantarius* LP90 revealed that it may be considered avirulent. The analysis of a 19,539-bp cluster showed the presence of 29 putative open reading frames (ORFs), including the genes encoding 8 new class II-bacteriocins, as well as the proteins involved in their secretion, immunity and regulation. Transcriptional analyses evidenced that the induction factor (IF) structural gene, the bacteriocin/IF transporter genes, the bacteriocin structural genes and most of the bacteriocin immunity genes were transcribed. MALDI-TOF analyses of peptides purified using different multichromatographic procedures revealed that the dairy strain *S. infantarius* LP90 produces at least 6 bacteriocins, including infantaricin $A_1$, a novel anti-pneumococcal two-peptide bacteriocin.

**Funding:** This work was supported by projects RTI2018-094907-B-I00 from Ministerio de Ciencia, Innovación y Universidades (MICINN, Madrid, Spain), 07CSA007E from the Junta de Galicia, Spain, AGL2012-34829 from Ministerio de Economía y Competitividad (MINECO, Madrid, Spain), and S-2009/AGR-1489 and S-2013/ABI-2747 from Consejería de Educación, Comunidad de Madrid, Madrid, Spain. C. Campanero and E. Muñoz-Atienza were recipients of predoctoral fellowships (FPU) from UCM, Spain. J. Feito holds a contract from UCM. S. Arbulu held a predoctoral fellowship (FPI) from MINECO.

**Competing interests:** The authors have declared that no competing interests exist.

## Introduction

*Streptococcus pneumoniae*, regarded as one of the most important human pathogens, is the etiological agent of diseases such as acute otitis media, sinusitis and pneumonia, as well as of potentially life-threatening invasive infections, such as bacteremia and meningitis, which constitute major causes of morbidity and mortality worldwide [1]. The development of vaccines could alleviate the problems related to the spreading of the most virulent strains. Nevertheless, several studies have revealed some drawbacks including the parallel increase of non-vaccine serotypes [2]. Although antibiotherapy constitutes a complementary strategy to vaccination, treatments with antibiotics may lead to dysbiosis and the spreading of antibiotic-resistant bacteria. The concerns about the worldwide emergence of antibiotic resistant *S. pneumoniae* strains, mainly against β-lactams and macrolides, and the raise of clinical isolates showing multidrug-resistance have increased [3]. Researchers are urged to discover and develop alternative or complementary strategies to antibiotherapy based on novel modes of action, such as bacteriocins produced by Lactic Acid Bacteria (LAB) [4]. Bacteriocins are ribosomally-synthesized antimicrobial peptides or proteins active against a wide variety of Gram-positive and, to a lesser extent, Gram-negative bacteria [5]. This work describes for the first time bacteriocin production by *Streptococcus infantarius*. The biochemical, genetic and transcriptional characterization of multibacteriocin production by the dairy strain *S. infantarius* LP90 revealed that it produces at least 6 bacteriocins, including infantaricin $A_1$, a novel anti-pneumococcal two-peptide bacteriocin which could be used as an alternative or complementary strategy to antibiotics for the treatment of human infections caused by *S. pneumoniae*, for which vaccination and antibiotic programs may be non-fully effective.

## Materials and methods

### Bacterial strains, growth media and culture conditions

The 37 bacteriocinogenic LAB and the 23 clinical *S. pneumoniae* strains used as indicators are listed in S1 Table. LAB were grown in de Man, Rogosa and Sharpe (MRS) broth (Oxoid Ltd., Basingstoke, United Kingdom) at 30˚C for 16 h. *S. pneumoniae* colonies were obtained on Columbia agar with horse blood (5%, w/v) (BioMérieux, Marcy l'Etoile, France) at 37˚C overnight. Subsequently, colonies were transferred to Brain Heart Infusion (BHI) broth (Oxoid), and cultures were incubated at 37˚C for approximately 6 h ($OD_{600}$ of 0.5).

### Detection of antimicrobial activity

**Direct antimicrobial activity assay.** The direct antimicrobial activity of the 37 bacteriocinogenic LAB was screened by a stab-on-agar test (SOAT) [6] against 9 *S. pneumoniae* strains (S2 Table). Briefly, the bacteriocinogenic strains were stabbed onto MRS agar (1.5% w/v) and incubated at 30˚C for 5 h, and then 40 ml of BHI soft (0.8%, w/v) agar (Oxoid) containing about $1 \times 10^5$ cfu/ml of the indicator strain was poured over the plates. After incubation, at 37˚C for 12 h, the plates were checked for inhibition zones around the colonies. Antimicrobial activity was quantified by measuring the diameter of the inhibition halo (mm), considering as positive results only diameters higher than 3 mm.

**Extracellular antimicrobial activity assay.** The extracellular antimicrobial activity activity of the 37 bacteriocinogenic LAB was determined by an agar well-diffusion test (ADT) [6] against 23. *S. pneumoniae* strains (S2 Table). Firstly, the cell-free culture supernatants (CFS) from the 37 bacteriocinogenic LAB grown in MRS broth at 30˚C until the stationary phase ($OD_{600}$ of ca. 1.0) were obtained by centrifugation at $12,000 \times g$ at 4˚C for 10 min. After adjusting their pH to 6.2 with 1M NaOH, they were filter-sterilized through 0.22 μm-pore-size filters

(Millipore Corporation, Bedford, Massachusetts, USA), and stored at –20˚C until use. Then, BHI soft agar (80 ml) were inoculated with approximately $1 \times 10^5$ cfu/ml of the *S. pneumoniae* indicator strains and poured into a Petri dish. After cooling, wells (6-mm diameter) were made in the agar, and 50 μl of CFS were added. After incubation of the plates at 37˚C for 12 h, the antimicrobial activity was quantified by measuring the diameter (mm) of the inhibition zones around the wells, and the bacteriocinogenic strains showing the highest antimicrobial activity were selected for further evaluation.

These strains were grown in MRS broth at 30 and 37˚C for 16 h and their CFSs obtained at 6, 10, 12, and 16 h were tested for antimicrobial activity by an ADT against 7 *S. pneumoniae* strains. CFSs showing the highest antimicrobial activity were challenged against the same strains by a microtiter plate assay (MPA) [7]. Each well of the microtiter plate contained 50 μl of two-fold serial dilutions (in BHI broth) of CFS and 150 μl of a diluted (in BHI broth) fresh overnight culture of the indicator (approx., $4 \times 10^5$ cfu/ml). After aerobic incubation of the plates at 37˚C for 12 h, growth inhibition was measured spectrophotometrically at 620 nm with a microtiter plate reader (Labsystems iEMS Reader MF, Labsystems, Helsinki, Finland). CFSs were heated at 100˚C for 10 min and treated with proteinase K (AppliChem GmbH, Darmstadt, Germany) (10 mg/ml) at 37˚C for 1 h.

## Taxonomic identification of *Streptococcus* sp. LP90

*Streptococcus* sp. LP90 was taxonomically identified by DNA sequencing of a PCR-amplified internal fragment of the manganese-dependent superoxide dismutase gene (*sodA*) using the primers d1 (5′-CCITAYICITAYGAYGCIYTIGARCC-3′) and d2 (5′-ARRTARTAIGC RTGYTCCCAIACRTC-3′) obtained from Sigma-Genosys Ltd. (Cambridge, United Kingdom) [8]. PCR mixtures were subjected to thermal cycling under the following conditions: initial denaturation (97˚C for 3 min) followed by 35 cycles of denaturation (94˚C for 30 sec), annealing (50˚C for 2 min), and elongation (72˚C for 1 min 30 sec), and a final 12-min extension at 72˚C, in an Eppendorf Mastercycler DNA thermal cycler (Eppendorf, Hamburg, Germany). The amplified PCR products were resolved by electrophoresis through agarose (2%, w/v) (Pronadisa, Madrid, Spain) gels stained with GelRed Nucleic Acid Gel Stain (Biotium, USA), and analyzed with the Gel Doc 1000 documentation system (Bio-Rad, Madrid, Spain). The molecular size marker used was HyperLadder II (Bioline London, United Kingdom). The PCR products were purified using the NucleoSpin Extract II (Macherey-Nagel, Düren, Germany) and sequenced at the Genomics Service of Parque Científico-UCM (Madrid, Spain). The nucleotide BLAST software (NCBI [http://blast.ncbi.nlm.nih.gov/Blast.cgi?PROGRAM= blastn&PAGE_TYPE=BlastSearch&LINK_LOC=blasthome]) was used for nucleotide similarity search.

## Safety assessment of *S. infantarius* LP90

**Antimicrobial susceptibility testing.** The minimum inhibitory concentration (MIC) of 16 antibiotics against *S. infantarius* LP90 was determined using Lact-1 and Lact-2 VetMIC microplates (National Veterinary Institute, Uppsala, Sweden) [9]. The antibiotics evaluated were ampicillin, penicillin, vancomycin, erythromycin, virginiamycin, tetracycline, clindamycin, chloramphenicol, kanamycin, gentamycin, streptomycin, neomycin, linezolid, rifampicin, ciprofloxacin, and trimethoprim. Individual colonies of *S. infantarius* LP90 were suspended in a sterile glass tube containing 5 ml of saline solution (0.85% NaCl, w/v) to a turbidity of 1 in the McFarland scale (approx., $3 \times 10^8$ cfu/ml) and further diluted 1,000-fold in susceptibility test medium (LSM medium) (Iso-Sensitest [IST]: MRS, 9:1) (Oxoid). Then, 100 μl of the diluted *S. infantarius* LP90 suspension was added to each microplate well, which was sealed

with a transparent covering tape and incubated at 37˚C for 18 h. After incubation, MICs were defined as the lowest concentration of the antibiotic that inhibited bacterial growth. Due to the lack of breakpoints for *S. infantarius*, the results were interpreted according to those established for *Streptococcus thermophilus* by the European Food Safety Authority (EFSA) [10]. *S. thermophilus* CECT801 was used as control.

**Phenotypic detection of virulence factors.**    Production of gelatinase, β-hemolysin and protease activity were determined on Todd Hewitt agar containing gelatine (3%, w/v) (Oxoid), Columbia agar supplemented with horse blood (5%, w/v), and Tryptone Soya agar (TSA) supplemented with skim milk (1.5%, w/v) (Oxoid), respectively. *E. faecalis* P4 and *E. faecalis* P36 [11] were used as positive controls. *S. infantarius* LP90 was grown in MRS broth at 37˚C in aerobic conditions for 16 h and then inoculated into MRS agar to obtain single colonies that were streaked onto the culture media cited above. After incubation at 37˚C for 24 h, the plates containing gelatine were placed at 4˚C for 5 h to facilitate the detection of zones of turbidity (indicating hydrolysis) around the colonies. The production of β-hemolysin and protease activity were detected by the presence of clear zones around the colonies.

**PCR-detection of genes encoding virulence factors.**    The presence of streptococcal and enterococcal virulence factor genes in *S. infantarius* LP90 was determined using the primer pairs listed in S3 Table. PCR mixtures were subjected to an initial cycle of denaturation (97˚C for 2 min), followed by 35 cycles of denaturation (94˚C for 45 sec), annealing (at an appropriate temperature [S3 Table] for 30 sec), and elongation (72˚C for 30 to 180 s), ending with a final extension step at 72˚C for 7 min. PCR products were resolved and analyzed as described above. The molecular size markers used were HyperLadder II (Bioline London, United Kingdom) and 1 Kb Plus DNA ladder (Invitrogen).

**Screening of histidine, tyrosine and ornithine decarboxylase activities in differential growth medium.**    Histidine, tyrosine and ornithine decarboxylase activities were evaluated using the improved decarboxylase differential growth medium [12]. *S. infantarius* LP90 was subcultured 5 times in MRS broth containing 0.1% (w/v) of the corresponding amino acid precursor (histidine, tyrosine or ornithine to check for HDC, TDC and ODC activities, respectively) (Sigma-Aldrich) and 0.005% (w/v) of pyridoxal-5-phosphate at 30˚C overnight. Subsequently, cultures were streaked in duplicate on decarboxylase differential growth medium agar plates with or without (negative control) the corresponding amino acid precursor, and incubated at 37˚C for 4 days under aerobic and anaerobic conditions. Biogenic amine production was detected by a color change from yellow to violet in the decarboxylase medium. *Lactobacillus brevis* CECT4121 was used as positive control for tyramine, and as negative control for histamine and putrescine. Likewise, *Lactobacillus* sp. ATCC30a was used as positive control for histamine and putrescine, and as negative control for tyramine [13].

**PCR-detection of histidine, tyrosine and ornithine decarboxylase genes.**    Total DNA from *S. infantarius* LP90 was subjected to PCR-amplification to detect histidine, tyrosine and ornithine decarboxylase genes (*hdc*, *tdc* and *odc*, respectively) using the primers CL1/JV17HC [14], TD2/TD5 [15] and 3/16 [16], respectively. *Lb*. *brevis* CECT4121 was used as positive control for *tdc*, and as negative control for *hdc* and *odc*. *Lactobacillus* sp. ATCC30a was used as positive control for *hdc* and *odc*, and as negative control for *tdc*. PCR mixtures prepared as described above were subjected to an initial cycle of denaturation (94–97˚C for 2 to 4 min), followed by 35 cycles of denaturation (94–95˚C for 45 to 60 s), annealing (48–52˚C for 30 sec), and elongation (72˚C for 60 to 180 s), ending with a final extension step at 72˚C for 7 min. PCR products were resolved and analyzed as described above.

**Degradation of mucin.**    The ability to degrade gastric mucin was determined as previously described [17]. Mucin from porcine stomach type III (Sigma-Aldrich) and agar were added to medium B (minimal anaerobic culture medium) without glucose at concentrations of 0.5 and

1.5% (w/v), respectively. A volume of 10 μl of an overnight liquid culture of *S. infantarius* LP90 was inoculated onto the surface of medium B. The plates were incubated anaerobically at 37˚C for 72 h, stained with amido black (0.1%, w/v) (Merck) in 3.5 M acetic acid (Merck) for 30 min, and then washed with 1.2 M acetic acid. A discolored zone around the spotted culture was considered as a positive result. A fresh fecal slurry from a healthy adult horse was used as positive control.

**Deconjugation of bile salts.** The ability to deconjugate bile salts was determined as previously described [18]. For this purpose, plates were prepared by adding sodium salts of taurocholate (TC) and taurodeoxycholate (TDC) (0.5%, w/v) (Sigma-Aldrich) to MRS agar supplemented with L-cysteine (0.05%, w/v) (Merck). A volume of 10μl of an overnight culture of *S. infantarius* LP90 was spotted onto the MRS agar plates supplemented with the bile salts, and the plates were incubated anaerobically at 37˚C for 72 h. The presence of precipitated bile acid around the spotted cultures was considered as a positive result. A fresh fecal slurry from a healthy adult horse was used as positive control.

**Determination of other enzymatic activities.** The APIZYM kit (BioMérieux) was used to determine the presence of the following enzymatic activities in *S. infantarius* LP90: esterase (C4), esterase lipase (C8), trypsin, α-chymotrypsin, leucine arylamidase, valine arylamidase, cystine arylamidase, acid phosphatase, naphthol-AS-BI phosphohydrolase, alkakine phosphatase, α-galactosidase, β-galactosidase, α-glucosidase, β-glucosidase, N-acetyl-β-glucosaminidase, β-glucuronidase, α-mannosidase, and α-fucosidase. Cells from an overnight *S. infantarius* LP90 culture were harvested by centrifugation at 12,000 × *g* for 2 min, resuspended in 2 ml of API Suspension Medium (BioMérieux) and adjusted to a turbidity of 5–6 in the McFarland scale (approx., $1.5–1.9×10^9$ cfu/ml). Aliquots of 65 μl of the suspensions were added to each of the 20 reaction cupules in the APIZYM strip. The strips were incubated at 37˚C for 4.5 h and the reactions were developed by the addition of one drop of each of the API-ZYM reagents A and B.

## PCR-detection of streptococcal and enterococcal bacteriocin structural genes in *S. infantarius* LP90

The presence of streptococcal and enterococcal bacteriocin structural genes in *S. infantarius* LP90 was determined by PCR using the primer pairs listed in S4 Table. PCR mixtures were subjected to an initial cycle of denaturation (97˚C for 2 min), followed by 35 cycles of denaturation (94˚C for 45 sec), annealing (at an appropriate temperature for 30 sec) and elongation (72˚C for 30 to 1 min 45 sec), ending with a final extension step at 72˚C for 7 min. PCR products were resolved and analyzed as described above.

## Purification, mass spectrometry and amino acid sequencing of the bacteriocins secreted by *S. infantarius* LP90

Bacteriocins secreted by *S. infantarius* LP90 were purified using modifications of the multichromatographic procedure described by Cintas *et al.* [6]. An overnight preculture of this strain was used to inoculate three 1 liter MRS broth bottles which were incubated at 37˚C under aerobic conditions for 12 h. After the cells were removed by centrifugation, the proteins present in the CFSs were subjected to precipitation with ammonium sulphate (50% [w/v] [Merck, Farma y Química S. A., Barcelona, Spain]). The solution was then stirred at 4˚C for 3 h and subsequently centrifuged (12,000 × *g* at 4˚C for 30 min). The pellet and floating materials obtained after centrifugation were mixed and solubilized in 200 ml of 20 mM sodium phosphate buffer (pH 6.0) and desalted by gel filtration (PD-10 columns; GE Healthcare Life Sciences, Barcelona, Spain). From this point, each of the three concentrated and desalted CFSs

was subjected to a different procedure: (i) cation-exchange (HiPrep 16/10 SP XL, GE Healthcare) and reversed-phase (PepRPC HR 5/5, GE Healthcare) chromatographies, both performed in a Fast Protein Liquid Chromatography system (ÄKTA-FPLC) (GE Healthcare) (procedure P1); (ii) anion-exchange (CM Sepharose Fast Flow, GE Healthcare) and hydrophobic-interaction (Octyl Sepharose CL-4B, Sigma-Aldrich) batch chromatographies, followed by reversed-phase chromatography in a ÄKTA-FPLC system (procedure P2), and (iii) cation-exchange (SP Sepharose Fast Flow, GE Healthcare) and hydrophobic-interaction batch chromatographies (procedure P3).

The antimicrobial activity of the fractions was determined by an MPA against *S. pneumoniae* 66405 and *Pediococcus damnosus* CECT4797. Purified peptides were subjected to mass spectrometry analysis in a Matrix-Assisted Laser Desorption Ionization-Time Of Flight Voyager-DE STR mass spectrometer (MALDI-TOF MS) (PerSeptive Biosystems, Foster City, California, USA) at the Mass Spectrum Service of Parque Científico-UCM. N-terminal amino acid sequencing of the purified anti-pneumococcal peptide was performed by Edman degradation using an ABI 494 Protein Sequencing System (Applied Biosystems, Foster City, California USA) at the Protein Chemistry Service of the Centro de Investigaciones Biológicas (CIB, Consejo Superior de Investigaciones Científicas, Madrid, Spain). The protein BLAST software (National Center for Biotechnological Information [NCBI], US National Library of Medicine, http://blast.ncbi.nlm.nih.gov) was used for amino acid similarity search. The N-terminal extensions of the putative antimicrobial peptides and the transmembrane segments of their putative immunity proteins were identified by the ExPASy (https://www.expasy.org/proteomics) and SOSUI (http://harrier.nagahama-i-bio.ac.jp/sosui) programs, respectively.

## Nucleotide sequencing of a multibacteriocinogenic gene cluster from *S. infantarius* LP90

Total DNA from *S. infantarius* LP90 was isolated using the Wizard Genomic DNA Purification kit (Promega). Specific oligonucleotide primers (Sigma-Genosys Ltd) designed according to the nucleotide sequence of *S. infantarius* ATCC BAA-102 (Genbank accession no. ABJK02000019) and to that of *S. infantarius* LP90 progressively determined by chromosome walking were used to obtain the sequence of the multibacteriocinogenic cluster. PCR mixtures were subjected to an initial cycle of denaturation (94˚C for 2 min), followed by 35 cycles of denaturation (94˚C for 15 sec), annealing (at an appropriate temperature for 30 sec), and elongation (68˚C for 30 sec to 1 min), ending with a final extension step at 68˚C for 7 min. The resulting products were resolved and analyzed as described above. The PCR-generated fragments were purified with the Nucleospin Extract II kit and sequenced at the Genomics Service of Parque Científico-UCM. Nucleotide sequence analysis was performed using the ORF-finder (NCBI) and the BLAST tools for the prediction of ORFs and similarity searches, respectively. The identification of *rho*-independent transcription terminators was performed using the ARNold finding terminator tool (Institut de Génétique et Microbiologie, Université Paris-Sud11 [http://rna.igmors.u-psud.fr/toolbox/arnold]). The 19,539 bp sequence obtained was deposited in GenBank under accession no. KJ475434.

## Transcriptional analysis of a multibacteriocinogenic cluster from *S. infantarius* LP90

Total RNA was isolated from exponentially growing *S. infantarius* LP90 cultures (OD$_{600}$ of 0.6–0.9) using the RNeasy Mini kit (Qiagen, Hilden, Germany), according to the procedure recommended by the manufacturer. The isolated RNA was treated with RNase-free DNase I (Qiagen) to thoroughly remove residual DNA. One-step reverse transcription (RT)-PCR was

performed using the Access RT-PCR System (Promega) with DNase-treated RNA as template for cDNA synthesis. Specific oligonucleotide primers, obtained from Sigma-Genosys Ltd., were designed according to the localization of promoter and transcription termination regions found in the nucleotide sequence of *S. infantarius* LP90 (19,539 bp) [region I (coordinates 1–3,878), sub-region IIa (coordinates 3,886–6,776), sub-region IIb (coordinates 6,829–7,877), sub-region IIc (coordinates 7,891–9,746), sub-region IId (coordinates 9,764–11,195), sub-region IIe (coordinates 11,212–11,953), and region III (coordinates 15,197–17,625). PCR amplifications using DNase-treated RNA as template were also performed as negative control. RT-PCR and PCR amplifications were carried out as recommended by the manufacturer (Promega). The resulting products were resolved and analyzed as described above.

## Results

### Antimicrobial activity of bacteriocinogenic LAB against clinical *S. pneumoniae* strains

The 37 bacteriocinogenic LAB, isolated in previously reported works (S1 Table), displayed a broad direct antimicrobial activity spectrum, inhibiting the growth of most of the 9 tested clinical *S. pneumoniae* strains (S2 Table). Thirty-one strains (84.2%) exhibited extracellular antimicrobial activity against, at least, one of the 23 tested clinical *S. pneumoniae* strains. The 6 bacteriocinogenic LAB showing the broadest spectrum and strongest extracellular antimicrobial activity were *Streptococcus* sp. LP90, *L. lactis* BB24, *E. faecalis* DBH18, *E. faecium* BNM58, *L. garvieae* DCC43 and *E. faecium* L50. The anti-pneumococcal activity of these strains withstood heat treatment at 100˚C for 1 h and was sensitive to proteinase K, indicating the proteinaceous nature of these antimicrobial compounds (*i.e.*, bacteriocins). These bacteriocinogenic strains were grown in MRS broth at 30 and 37˚C for 16 h to ascertain the best conditions for maximum bacteriocin production (results not shown). *Streptococcus* sp. LP90 grown at 37˚C for 12 h displayed the strongest extracellular antimicrobial activity and the broadest anti-pneumococcal spectrum inhibiting the growth of the 7 *S. pneumoniae* strains used as indicators (results not shown). This strain was taxonomically identified as *Streptococcus infantarius* subsp. *infantarius* and selected for further studies.

### Safety assessment of *S. infantarius* LP90

*S. infantarius* LP90 was sensitive to all the tested antibiotics for which a breakpoint has been established by EFSA (Table 1). *S. infantarius* LP90 showed no hemolysin, gelatinase and protease activities (S1 Fig). None of the tested genes encoding streptococcal and enterococcal virulence factors were detected in this strain (results not shown). Histidine, tyrosine and ornithine decarboxylase activities (S2 Fig) and the respective genes *hdc*, *tdc* and *odc* were not detected in *S. infantarius* LP90 (results not shown). Only the peptidase activities leucine and cystine arylamidases were found (results not shown). Mucine degradation and bile salt deconjugation abilities were not detected in *S. infantarius* LP90 (S3 Fig).

### PCR-detection of streptococcal and enterococcal bacteriocin structural genes in *S. infantarius* LP90

None of the tested streptococcal and enterococcal bacteriocin genes were amplified by PCR in this bacteriocinogenic strain (results not shown).

**Table 1. Antibiotic susceptibility of *S. infantarius* LP90[a].**

| Antibiotic | | | MICs (µg/ml)[b] | FEEDAP BPs values[c, d] | Susceptibility/ Resistance[d] |
|---|---|---|---|---|---|
| Mechanism of action/ Family | | Test range (µg/ml) | | | |
| **Cell wall synthesis inhibitors** | | | | | |
| Penicillins | Ampicillin | 0.03–16 | 0,12 | 2 | S |
| | Penicillin | 0.03–16 | ≤0.06 | Nav | Na |
| Glycopeptides | Vancomycin | 0.25–128 | 0,50 | 4 | S |
| **Protein synthesis inhibitors** | | | | | |
| Macrolides | Erythromycin | 0.016–8 | 0,12 | 2 | S |
| Streptogramins | Virginiamycin | 0.016–8 | ≤0.25 | Nav | Na |
| Tetracyclines | Tetracycline | 0.12–64 | 0,25 | 4 | S |
| Lincosamides | Clindamycin | 0.03–16 | 0,06 | 2 | S |
| Amphenicols | Chloramphenicol | 0.12–64 | 2,00 | 4 | S |
| Aminoglycosides | Kanamycin | 2–1024 | 64,00 | 64 | S |
| | Gentamycin | 0.5–256 | 2,00 | 32 | S |
| | Streptomycin | 0.5–256 | 32,00 | 64 | S |
| | Neomycin | 0.5–256 | ≤8.00 | Nav | Na |
| Oxazolidinone | Linezolid | 0.25–16 | ≤1.00 | Nav | Na |
| **Nucleic acid synthesis inhibitors** | | | | | |
| Rifamycins | Rifampicin | 0.12–64 | ≤0.50 | Nav | Na |
| Fluoroquinolones | Ciprofloxacin | 0.25–128 | ≤2.00 | Nav | Na |
| Trimethoprim | Trimethoprim | 0.12–64 | ≤32.00 | Nav | Na |

[a]Antibiotic susceptibility determined by a broth microdilution test. MICs were defined as the lowest concentration of the antibiotic at which no growth was detected.

[b]MICs determined by a VetMIC test

[c]Breakpoints (BPs) for *S. thermophilus* proposed by EFSA [10]. BPs for penicillin, virginiamycin, neomycin, linezolid, rifampicin, ciprofloxacin, and trimethoprim are not available (Nav).

[d]Based on the EFSA [10] BPs, *S. infantarius* LP90 was categorized as susceptible (S) or resistant (R) to the corresponding antibiotic. Na, not applicable.

## Purification, mass spectrometry and amino acid sequencing of the anti-pneumococcal bacteriocin secreted by *S. infantarius* LP90

Purification of the anti-pneumococcal bacteriocin produced by *S. infantarius* LP90 revealed a single well-separated absorbance peak with antimicrobial activity which eluted at 32% (v/v) 2-propanol in aqueous 0.1% (v/v) TFA (results not shown). The purity and molecular mass of the purified bacteriocin was determined by MALDI-TOF MS, which revealed a major peptide with a molecular mass of 3,963 Da (Fig 1). An N-terminal amino acid sequence of 19 residues of this peptide was obtained by Edman degradation (Fig 2). This anti-pneumococcal bacteriocin, which was termed infantaricin $A_1$ (InfA$_1$), is highly similar (74% of identity) to a "bacteriocin-type signal sequence" from *S. infantarius* subsp. *infantarius* ATCCBAA-102 deposited in the Genbank database (accession no. EDT47634.1) (Fig 2).

## Nucleotide sequence and organization of a multibacteriocinogenic gene cluster from *S. infantarius* LP90

The 19,539-bp sequence of *S. infantarius* LP90 (accession no. KJ475434) shows a 99% identity to a nucleotide sequence of *S. infantarius* ATCCBAA-102, and includes 29 putative ORFs (*orf1-orf29*) (Table 2 and Fig 3) located on the same DNA strand (except *orf1*, *orf2*, and *orf29*) and preceded by putative ribosome binding sites (RBSs), except *orf14* and *orf21*. These ORFs can be grouped into three functional regions (Fig 3): (i) region I (*orf1-orf2*), related to

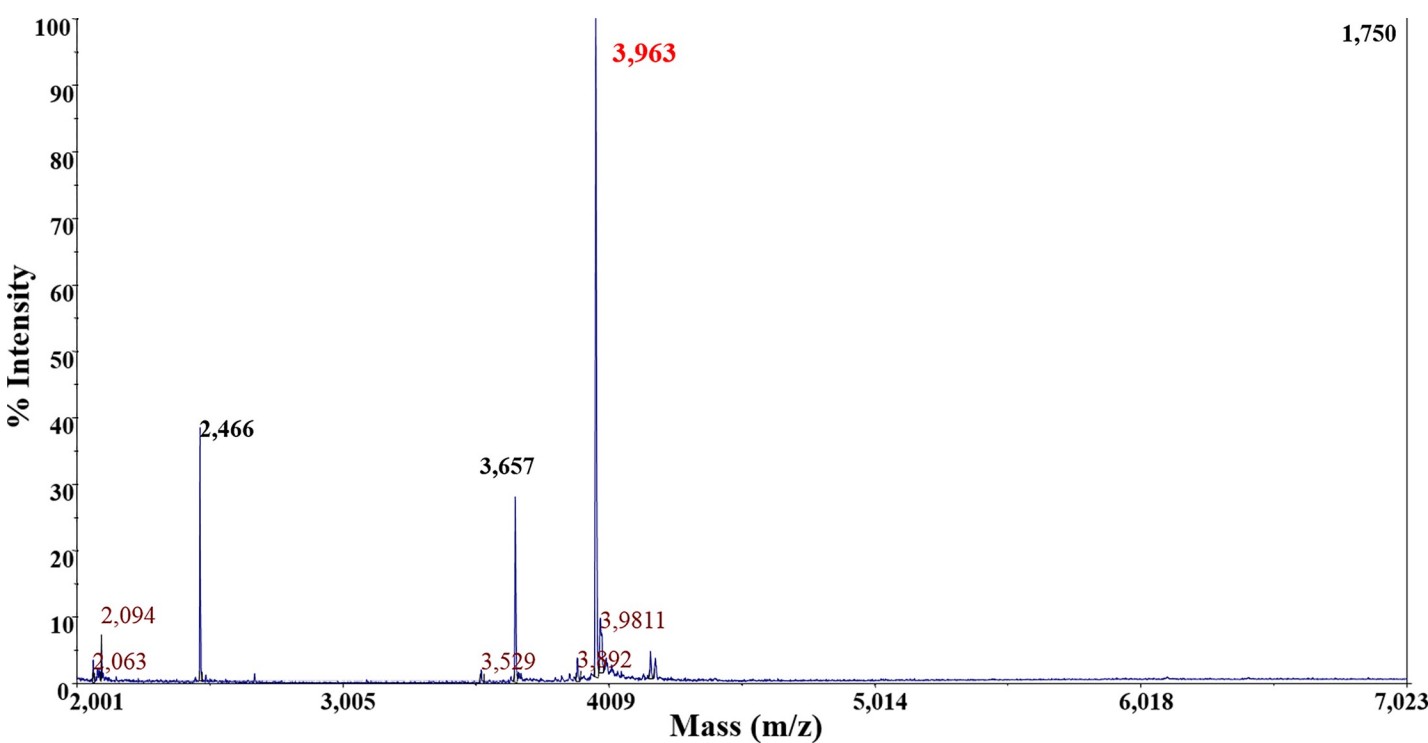

**Fig 1. Mass spectrometry analysis of the purified anti-pneumococcal bacteriocin infantaricin A₁ secreted by *S. infantarius* LP90.**

bacteriocin/IF secretion and organized in one operon; (ii) region II (*orf3-orf26*) involved in bacteriocin synthesis and immunity and organized in seven operons (sub-divided into six sub-regions, IIa to IIf), and (iii) region III (*orf27-orf29*), related to regulation of the bacteriocin production and organized in two operons. The region II (sub-regions IIa, IIc, and IId) includes 13 genes likely encoding bacteriocins according to their homology to known bacteriocins (Table 2), physico-chemical characteristics (Table 3) and the presence of N-terminal extensions of the GG-type of 15–30 amino acid residues (results not shown). The proteins encoded by *orf9*, *orf10*, *orf25* and *orf26* contain six, three, three and two transmembrane segments, respectively, and *orf11* and *orf16* seem to encode soluble proteins (results not shown).

## Transcriptional analysis of a multibacteriocinogenic cluster from *S. infantarius* LP90

mRNA transcripts of the expected sizes were detected in the following regions (Fig 3): (i) region I (putative transcript T1, PT1) (*orf1-orf2*); (ii) region II, sub-region IIa (PT2, *orf3-orf9*), sub-region IIb (PT3, *orf10-orf11*), sub-region IIc ([PT4, *orf12-orf14*] and [PT5, *orf15-orf16*]),

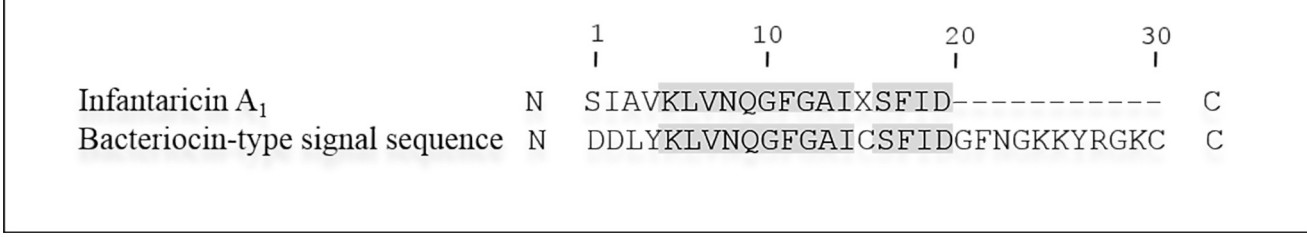

**Fig 2. Alignment of the partial N-terminal sequence of infantaricin A₁ and the bacteriocin-type signal sequence from *S. infantarius* subsp. *infantarius* ATCCBAA-102.** Identical residues are shaded in grey. "X" indicates an unidentified residue.

**Table 2. Predicted Open Reading Frames (ORFs) in the 19,539-bp fragment of genome from *S. infantarius* LP90 and the closest relationships of the deduced proteins.**

| ORF | Gene | | | Protein | | | |
|-----|------|------|------|---------|---|---|---|
| | Position in nucleotide sequence | | Name | Length (amino acids) | Molecular mass (kDa) | pI | Closest relationship[a] (number of amino acid; E-value, level of amino acid identity/ level of amino acid similarity; microorganism) (Accession no.) |
| | 5' | 3' | | | | | |
| ORF1 | 1.563 | 169 | orf1 | 464 | 51.4 | 5.90 | Bacteriocin secretion accesory protein (464; 0.0, 99%/99%; *S. gallolyticus*) (WP_074658230.1) |
| ORF2 | 3.722 | 1.574 | orf2 | 715 | 80.1 | 6.70 | Peptide cleavage/export ABC transporter ATP (715; 0.0, 99%/99%; *S. infantarius*) (WP_043878344.1) |
| ORF3 | 3.997 | 4.254 | orf3 (infC$_2$) | 85 | 8.2 | 4.60 | Bacteriocin BlpJ (85; 1e-15, 98%/98%; *S. infantarius* ATCC BAA-102) (EDT47628.1) |
| ORF4 | 4.272 | 4.466 | orf4 (infC$_1$) | 64 | 6.0 | 4.50 | Class IIb bacteriocin, lactobinA/cerein7B family (71; 2e-21, 100%/100%; *S. infantarius* subsp. *infantarius* ATCC BAA-102) (EDT47629.1) |
| ORF5 | 4.662 | 4.889 | orf5 (infB$_2$) | 75 | 7.2 | 4.00 | Bacteriocin BlpJ (85; 1e-05, 83%/86%; *S. infantarius* ATCC BAA-102) (EDT47628.1) |
| ORF6 | 4.873 | 5.079 | orf6 (infB$_1$) | 68 | 6.9 | 4.30 | Class IIb bacteriocin, lactobin A/cerein 7B family (68; 1e-23, 99%/98%; *Streptococcus caballi*) (WP_040807669.1) |
| ORF7 | 5.548 | 5.706 | orf7 (infA$_2$) | 52 | 5.5 | 9.80 | Hypothetical protein (52; 7e-28; 100%/100%; *S. infantarius* ATCC BAA-102) (EDT47633.1) |
| ORF8 | 5.734 | 5.886 | orf8 (infA$_1$) | 51 | 5.8 | 7.70 | Bacteriocin (54; 2e-08, 53%/70%; *Streptococcus thoraltensis*) (WP_018380431.1) |
| ORF9 | 5.929 | 6.604 | orf9 | 224 | 25.2 | 9.30 | CAAX amino terminal protease family membrane protein (224; 3e-129, 100%/100%; *S. infantarius*) (SUN69110.1) |
| ORF10 | 6.944 | 7.246 | orf10 | 100 | 11.4 | 9.20 | Putative bacteriocin self-immunity protein (100; 1e-55, 100%/100%; *S. infantarius*) (SUN69109.1) |
| ORF11 | 7,540 | 7.842 | orf11 | 100 | 11.6 | 7.90 | Enterocin A immunity protein (100; 8e-64, 100%/100%; *S. infantarius*) (SUN69108.1) |
| ORF12 | 8,000 | 8.233 | orf12 (infD$_1$) | 77 | 7.5 | 4.90 | Bacteriocin-like peptide BlpJ (77; 1e-45, 100%/100%; *S. infantarius*) (SUN69107.1) |
| ORF13 | 8.253 | 8.441 | orf13 (infD$_2$) | 62 | 6.0 | 4.70 | Class IIb bacteriocin, lactobin A/cerein 7B family (41; 0.075, 100%/100%; *S. infantarius* ATCC BAA-102) (EDT47640.1) |
| ORF14 | 8.721 | 8.936 | orf14 | 71 | 8.4 | 9.10 | Hypothetical protein (71; 4e-12, 94%/98%; *S. gallolyticus* subsp. *gallolyticus* TX20005) (EFM28808.1) |
| ORF15 | 9.191 | 9.409 | orf15 (infE) | 72 | 7.7 | 4.90 | Bacteriocin piscicolin-126 (72; 5e-41, 93%/95%; *Streptococcus equinus*ATCC 700338) (EFM26697.1) |
| ORF16 | 9.409 | 9.705 | orf16 | 98 | 11.3 | 9.40 | Bacteriocin immunity protein (98; 9e-62, 99%/100%; *S. infantarius* CJ18) (AEZ63022.1) |
| ORF17 | 9.874 | 10.107 | orf17 (infF) | 77 | 7.8 | 7.70 | Bacteriocin BlpK (77; 2e-29, 100%/100%; *S. gallolyticus*) (ALT80184.1) |
| ORF18 | 10.112 | 10.282 | orf18 | 56 | 6.8 | 9.80 | Hypothetical protein (65; 2e-21, 98%/100%; *S. caballi*) (WP_018364974.1) |
| ORF19 | 10.297 | 10.497 | orf19 (infG$_1$) | 66 | 6.5 | 6.00 | Bacteriocin class II (66; 2e-18, 98%/98%; *S. infantarius* CJ18) (AEZ63021.1) |
| ORF20 | 10.523 | 10.705 | orf20 (infG$_2$) | 60 | 5.8 | 4.30 | Bacteriocin class II (60; 9e-14, 100%/100%; *S. infantarius* CJ18) (AEZ63020.1) |
| ORF21 | 10.677 | 10.829 | orf21 | 50 | 5.9 | 9.52 | Hypothetical protein (48; 6e-21, 88%/93%; *S. gallolyticus*TX20005) (EFM28809.1) |
| ORF22 | 10.854 | 11.069 | orf22 | 71 | 8.3 | 8.80 | Hypothetical protein (71; 2e-15, 100%/100%; *S. gallolyticus*) (SEF21267.1) |
| ORF23 | 11.323 | 11.553 | orf 23 (infH) | 76 | 8.0 | 5.64 | Bacteriocin class II with double-glycine leader peptide (76;6e-49,100%/100%; *S. infantarius*) (SUN69099.1) |
| ORF24 | 11.642 | 11.951 | orf24 | 103 | 12.0 | 9.30 | Hypothetical protein (103; 9e-69, 98%/99%; *S. equinus*) (EFM26700.1) |

*(Continued)*

**Table 2.** (Continued)

| ORF | Gene | | | Protein | | | |
|---|---|---|---|---|---|---|---|
| | Position in nucleotide sequence | | Name | Length (amino acids) | Molecular mass (kDa) | pI | Closest relationship[a] (number of amino acid; E-value, level of amino acid identity/ level of amino acid similarity; microorganism) (Accession no.) |
| | 5' | 3' | | | | | |
| ORF25 | 12.367 | 12,720 | *orf25* | 117 | 13.0 | 8.79 | Immunity protein BlpL (134; 6e-62,99%/99%; *S. infantarius* CJ18) |
| | | | | | | | (AEZ63016.1) |
| ORF26 | 12.901 | 13.947 | *orf26* | 348 | 37.6 | 4.16 | LPXTG cell wall anchor domain-containing protein (348; 0.0, 100%/100%; |
| | | | | | | | *S. infantarius*) (WP_006531888.1) |
| ORF27 | 15.304 | 16.029 | *orf27* | 241 | 27.9 | 5.25 | DNA-binding response regulator (241; 2e-177, 99%/100%; *Streptococcus* sp.) |
| | | | | | | | (HAK39972.1) |
| ORF28 | 16.042 | 17.351 | *orf28* | 439 | 51.4 | 5.85 | Histidine kinase (439; 0.0, 99%/99%; *Streptococcus* sp.) |
| | | | | | | | (HAK39971.1) |
| ORF29 | 17.521 | 17.373 | *orf29* | 21 | 2.6 | 6.90 | Protein of the ComC family (49; 5e-28, 100%/100%; *S. infantarius* CJ18) |
| | | | | | | | (AEZ63012.1) |

[a]Results of the similarity searches conducted on October of 2019.

sub-region IId (PT6, *orf17-orf22*), and sub-region IIe (PT7, *orf23-orf24*), and (iii) region III (PT10, *orf29*). No RT-PCR amplicons were detected at sub-region IIf (PT8) (*orf25* and *orf26*) and region III (PT9) (*orf27-orf28*). The following regions were found to be co-transcribed: (i) sub-regions IIa and IIb (PT2-3, *orf3-orf11*); (ii) sub-regions IIb and IIc (PT3-4, *orf10-orf14*); (iii) sub-region IIc (PT4-5, *orf12-orf16*), and (iv) sub-regions IId and IIe (PT6-7, *orf17-orf24*). No co-transcriptional unit was found between sub-regions IIc and IId (PT5-6, *orf15-orf22*).

## Purification and mass spectrometry of the non-anti-pneumococcal bacteriocins secreted by *S. infantarius* LP90

**Bacteriocin purification based on cation-exchange and RP-FPLC chromatographies (procedure P1).** RP-FPLC resulted in one well-separated absorbance peak eluting at 30% (v/

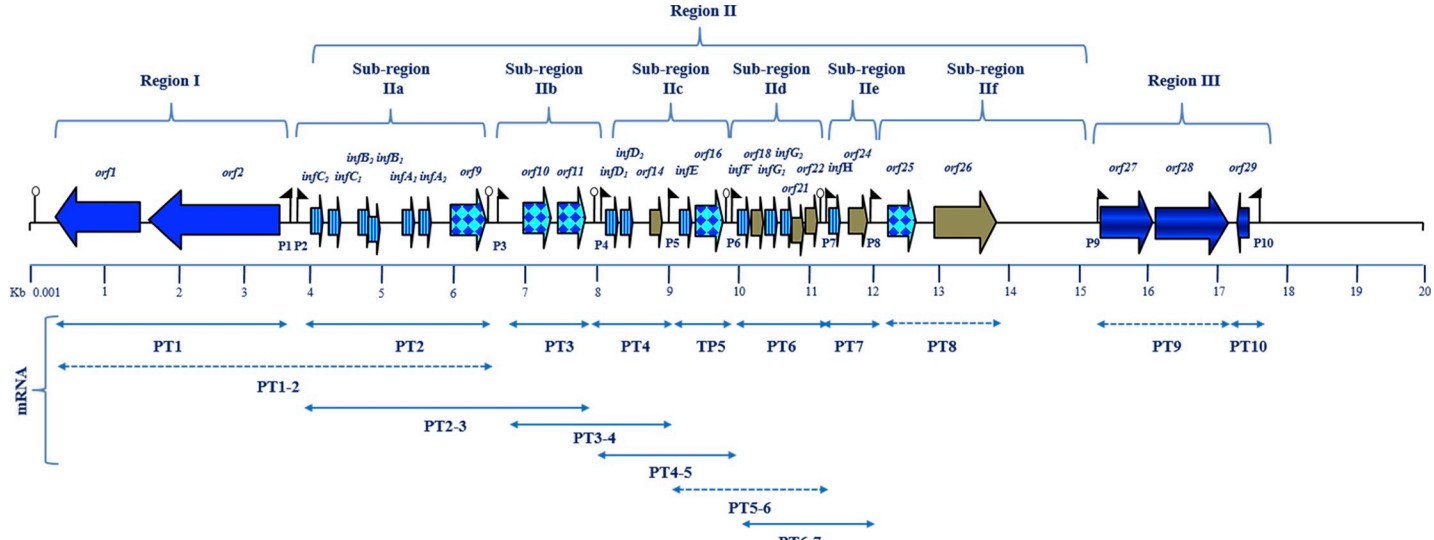

**Fig 3. Physical map showing the genetic organization of a 19,539-bp multibacteriocinogenic cluster present in *S. infantarius* LP90.** Putative ORFs are represented by wide arrows indicating the direction of transcription. Small flags and loop symbols indicate putative promoters and *rho*-independent transcription termination regions, respectively. Horizontal arrows below the genes refer to the transcriptional and co-transcriptional units as determined by RT-PCR. Dashed arrows refer to genes not transcribed or not co-transcribed.

**Table 3. Hypothetical bacteriocins encoded by *S. infantarius* LP90.**

| Gene | | Protein | | | | |
|------|------|------|------|------|------|------|
| ORF | Name | Name | Length | Size | pI | Amino acid sequence |
| | | | (aa) | (Da) | | |
| *orf3* | *infC$_2$* | InfC$_2$ | 62 | 5.680 | 8.05 | YSAIDCRNAMLTGIASGIIAGGTGAGLATLGVGGLAGAFVGAHIGAIGGGLTCVGGMIGNKF |
| *orf4* | *infC$_1$* | InfC$_1$ | 42 | 3.702 | 8.05 | KGNLGSAIGGCIGGVILAAATGPLTGGAAALVCVGSGLSAYL |
| *orf5* | *infB$_2$* | InfB$_2$ | 52 | 4.681 | 8.03 | YSAGACLTDIGLGMVGGGAAGVAGGVAGIALGVALGQIGGNVRCITSYVGGK |
| *orf6* | *infB$_1$* | InfB$_1$ | 43 | 4.009 | 4.37 | VNWGVAAGYCVAGALIGAAGGAISAGVGCLVSGLQEVSDGLFK |
| *orf7* | *infA$_2$* | InfA$_2$ | 29 | 3.000 | 10.40 | DNITKGAYKLGKATRQAAKAIGKAVGKLF |
| *orf8* | *infA$_1$* | InfA$_1$ | 30 | 3.357 | 8.76 | DDLYKLVNQGFGAICSFIDGFNGKKYRGKC |
| *orf12* | *infD$_1$* | InfD$_1$ | 54 | 4.988 | 8.05 | NKCVNAIFGGALTGAGSGFVGGMATLGVTSIPGAFVGAHFGAIAGGLYCVGASL |
| *orf13* | *infD$_2$* | InfD$_2$ | 43 | 3.902 | 8.03 | GKVGAAVGGCLGGMLLAWAGGPVTATGYAVVCGTAGAAAAYFN |
| *orf15* | *infE* | InfE | 49 | 5.162 | 8.79 | KTIYYGNGLYCNANKCWVNWSQTATTIANNSVINGLTGGNAGWHSGGIA |
| *orf17* | *infF* | InfF | 54 | 5.290 | 9.70 | GCNWKGAAATVAVGAVGGAIKGAVTTYSWQGAALKAVGYGIKAGVAYGVTCRWT |
| *orf19* | *infG$_1$* | InfG$_1$ | 43 | 4.033 | 10.10 | KSKEGRNMGCILGTAGMAGAGFLVAGPAGAAALGGATALRVCR |
| *orf20* | *infG$_2$* | InfG$_2$ | 42 | 3.764 | 8.90 | KVSPTCAALVAASIYGGLAVAGPAGVGLAMAVGGAAAGSFCR |
| *orf23* | *infH* | InfH | 53 | 5.439 | 8.90 | GMEWIGDVLGAIGNAAHPVNPRQVVDQLNGKYPRRGPVRSCPPGGTGGTPNAC |

v) 2-propanol in aqueous TFA (0.1%, v/v) (results not shown) showing antimicrobial activity against *P. damnosus* CECT4797. MALDI-TOF MS of this fraction revealed three major peaks of 4,714; 3,354 and 2,359 Da (Fig 4).

**Bacteriocin purification based on anion-exchange and hydrophobic interaction batch chromatographies followed by RP-FPLC (procedure P2).** RP-FPLC resulted in three peaks (fractions A, B and C) eluting at 34, 36 and 38% (v/v) 2-propanol in aqueous TFA (0.1%, v/v), respectively (results not shown), showing antimicrobial activity against *P. damnosus* CECT4797. MALDI-TOF MS revealed predominant peaks of 4,718; 3,968 and 4,067 Da in fractions A (Fig 5A), fraction B (Fig 5B) and fraction C (Fig 5C), respectively.

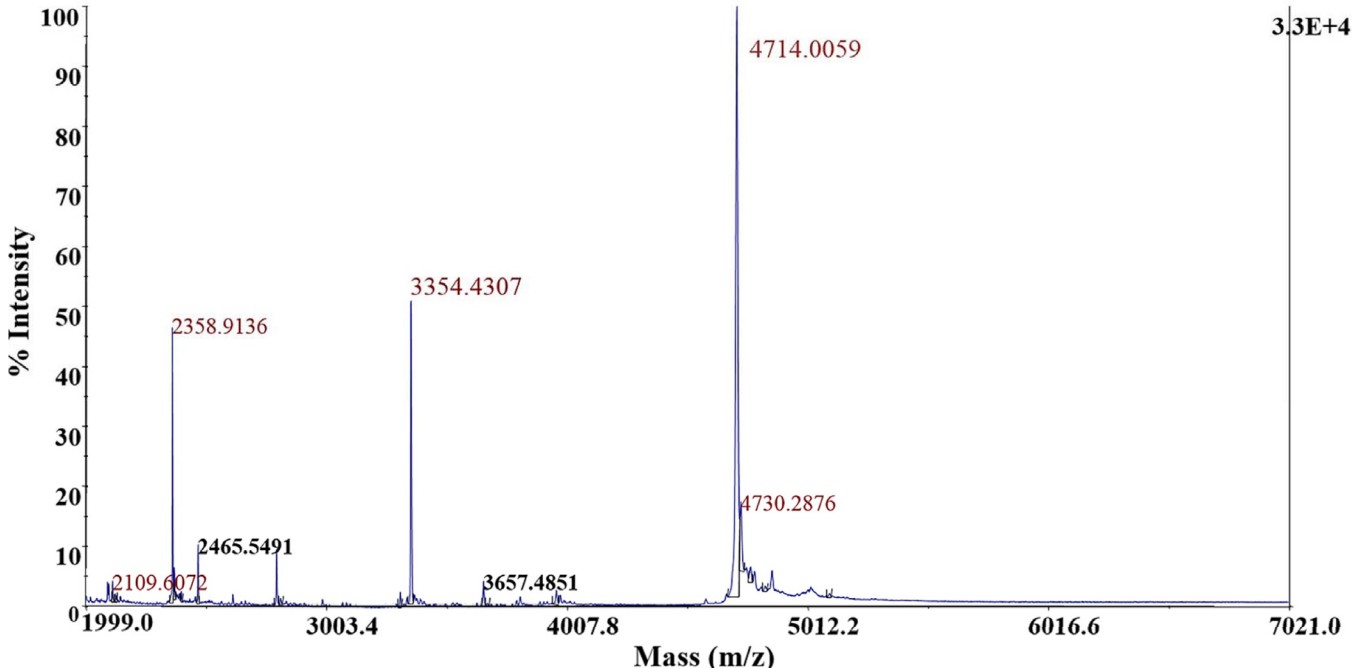

**Fig 4. Mass spectrometry analysis of the antimicrobial peptides obtained from *S. infantarius* LP90 using the purification procedure P1.**

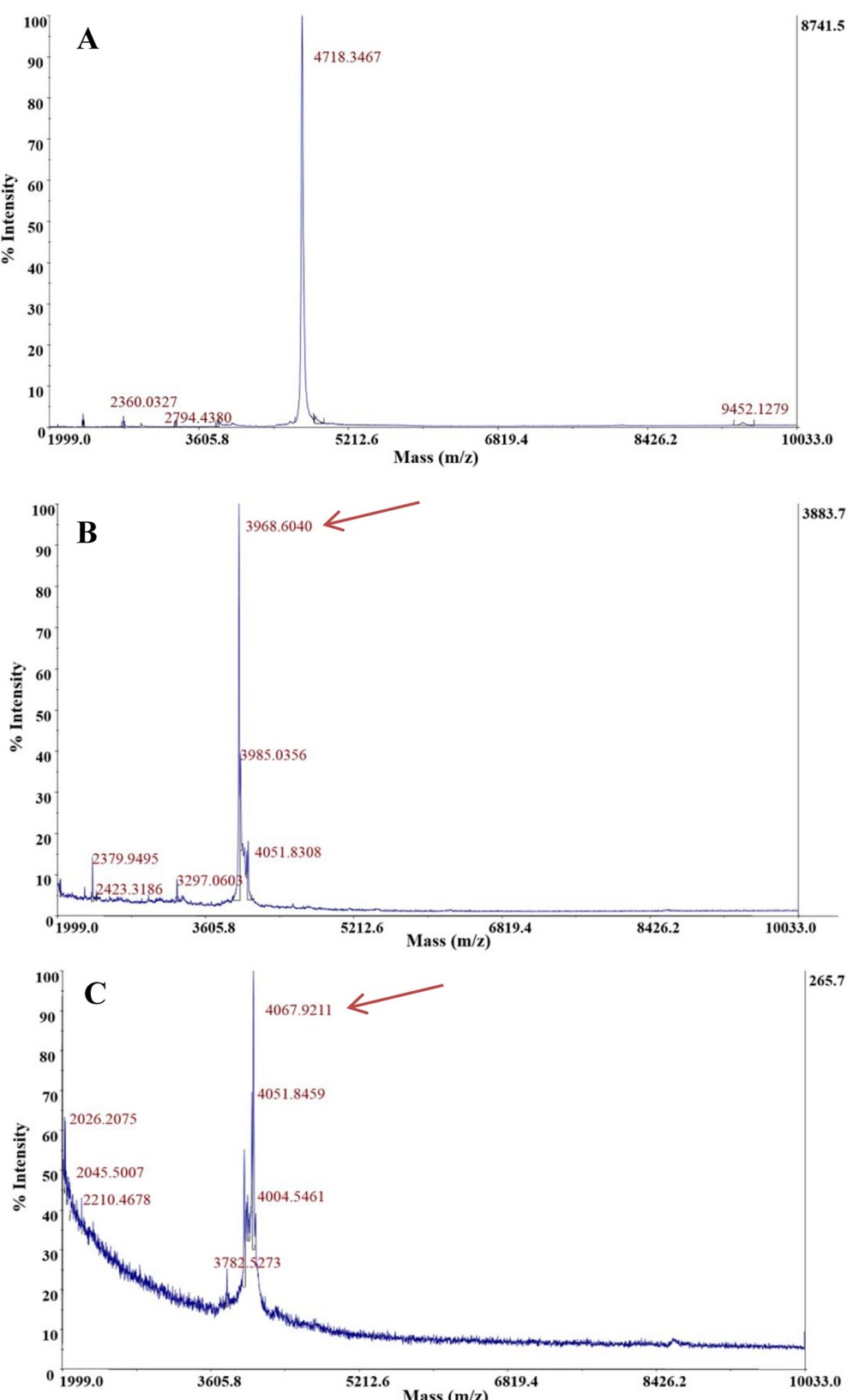

**Fig 5. Mass spectrometry analysis of the antimicrobial peptides obtained from *S. infantarius* LP90 using the purification procedure P2. a**. Molecular mass of the peak eluting at 34% (v/v) 2-propanol in aqueous TFA (0.1%, v/v). **b.** Molecular mass of the peak eluting at 36% (v/v) 2-propanol in aqueous TFA (0.1%, v/v). **c.** Molecular mass of the peak eluting at 38% (v/v) 2-propanol in aqueous TFA (0.1%, v/v). The predominant peak is pointed out with an arrow.

**Partial bacteriocin purification based on cation-exchange and hydrophobic-interaction batch chromatographies (procedure P3).**   MS-analysis of a crude fraction obtained after gel filtration, cation-exchange and hydrophobic-interaction batch chromatographies of the proteins present in a *S. infantarius* LP90 CFS resulted in several peaks with molecular masses included in the size range of bacteriocins (3.5–5.5 kDa), namely, 3,654; 3,709; 3,890; 4032; 4,718 and 5,439 Da (Fig 6A), and 3,963; 4,716 and 5,436 Da (Fig 6B).

## Discussion

Treatment of *S. pneumoniae* infections represents a challenge because of the emergence of multidrug-resistant pneumococci [3]. LAB bacteriocins might constitute an alternative or complementary treatment to antibiotics [4]. From 37 bacteriocinogenic LAB isolated from food and other sources, *S. infantarius* subsp. *infantarius* LP90, obtained from Venezuelan water buffalo milk [19], displayed the broadest and strongest anti-pneumococcal spectrum. *S. infantarius* belongs to the *Streptococcus bovis-Streptococcus equinus* group, comprising commensal non-β haemolytic inhabitants of the gastrointestinal tract of animals and humans [20]. This species has been isolated from dairy products [21] where it seems to fulfil the function of *S. thermophilus* [22], frozen peas, faeces of infants and clinical samples from patients suffering from endocarditis [23]. *S. infantarius* LP90 does not encode the glucosyltransferase enzyme,

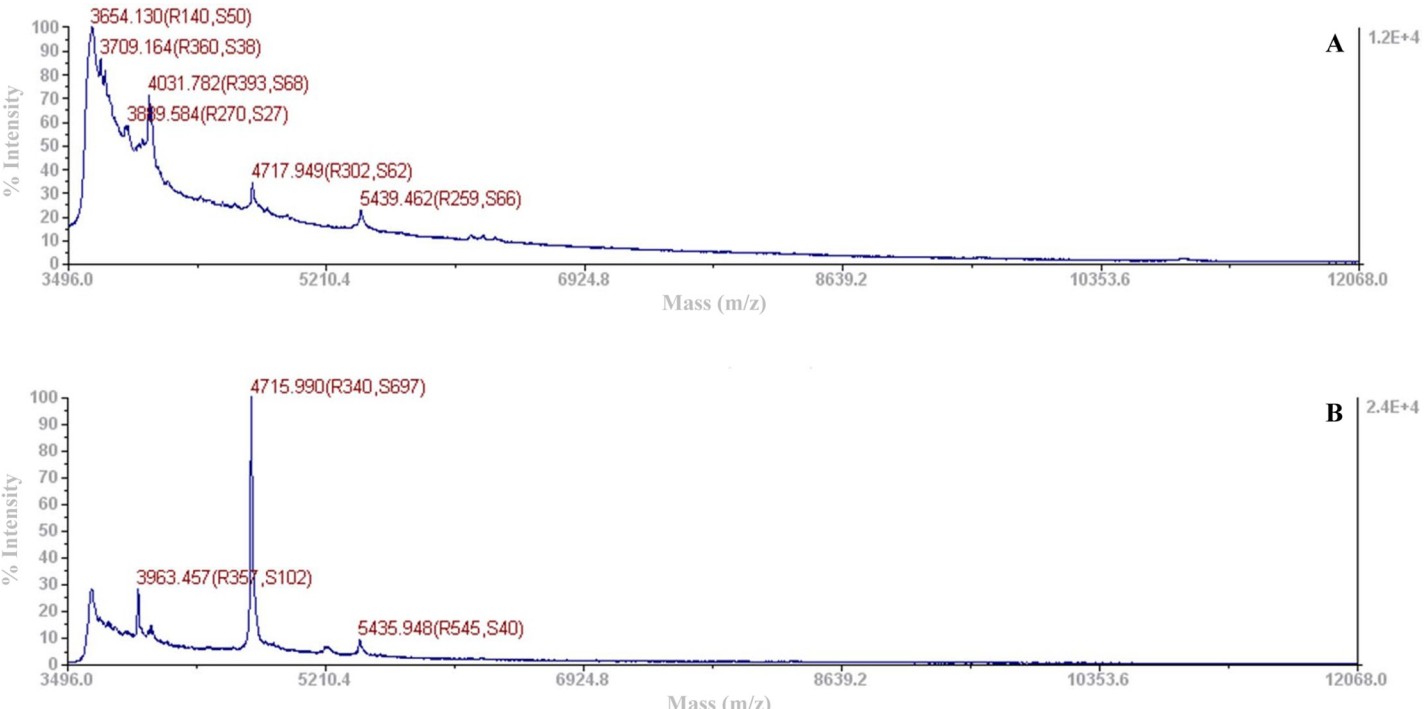

**Fig 6. Mass spectrometry analysis of the proteins obtained from *S. infantarius* LP90 using the purification procedure P3. a.** Molecular mass of the crude sample obtained after hydrophobic-interaction batch chromatography. **b.** Molecular mass of the precipitated sample (2-D Clean-Up Kit, Sigma-Aldrich) obtained after hydrophobic-interaction batch chromatography.

which might constitute a virulence factor for humans and animals [21], or any of the previously described streptococcal and enterococcal virulence factors; moreover, it does not produce any of the tested biogenic amines, nor exhibits β-glucuronidase, mucinolytic or bile salt deconjugation activities. *S. infantarius* LP90 shows an antibiotic susceptibility profile similar to *S. thermophilus*, a species used as dairy starter culture [24]. Therefore, this strain can be considered as safe.

Analysis of the 19,539-bp cluster from *S. infantarius* LP90 reveals that this is a multibacteriocinogenic producer whose bacteriocins would share the same secretion and regulation system. Region II contains 13 genes (*orf3-orf8*, *orf12-orf13*, *orf15*, *orf17*, *orf19-orf20* and *orf23*) encoding bacteriocins (Table 2). These peptides have a relatively low size (<6 kDa) and a high pI (Table 3), which are characteristic of class I and II bacteriocins, and contain N-terminal extensions of the GG-type of 15–30 amino acid residues, which are typical of bacteriocins secreted by dedicated ABC transporters [25]. Ten out of these 13 bacteriocin structural genes might act synergistically (subclass IIb bacteriocins), as suggested by their tandem organization and high homology to other two-peptide bacteriocins, thus constituting the two-peptide bacteriocins InfC, InfB, InfA, InfD and InfG. On the other hand, InfE is a subclass IIa bacteriocin, while InfF and InfH are subclass IIe bacteriocins. Several loci displaying characteristics of class II bacteriocin genes have been identified in the genome of different *Streptococcus* spp. [26]. Moreover, the presence of a high number of bacteriocin genes, although lower than that found in this study, seems to be common in streptococci [26]. Region II also contains 4 genes displaying homology to bacteriocin immunity proteins, and 1 gene showing homology to CAAX amino-terminal proteases (Abi proteins), which have been shown to be involved in self-immunity against bacteriocins [27]. RT-PCR showed that bacteriocin/IF transporter genes (region I) and all the bacteriocin structural and immunity genes (region II), except *orf25* and *orf26*, are transcribed. Amongst the bacteriocin regulation genes (region III), only *orf29*, which based on the sequence similarity analysis likely encodes an IF, was transcribed.

The 3,963 and 3,354 Da peptides secreted by *S. infantarius* LP90 could be two independent peptides or, more likely, two $InfA_1$ variants. $InfA_1$ (3,963 Da) shows anti-pneumococcal activity but its molecular mass does not match with the predicted one (3,357 Da), while $InfA_1$ with the predicted molecular mass (3,354 Da, Fig 4) does not show inhibitory activity against *S. pneumoniae*. From these observations, three hypotheses are proposed: (i) $InfA_1$ (3,963 Da) is somehow modified or bounded to an unknown molecule which confers anti-pneumococcal activity; (ii) the amount of $InfA_1$ (3,357 Da) produced by *S. infantarius* LP90 is not high enough to inhibit *S. pneumoniae*, while the amount of $InfA_1$ (3,963 Da) would be higher and thus it would permit to detect its anti-pneumococcal activity (in this case, the hypothetical modification of $InfA_1$ would not be positively involved in the anti-pneumococcal activity), and (iii) considering that $InfA_1$ and $InfA_2$ seem to constitute a two-peptide bacteriocin, $InfA_2$ could be present in the same fraction as $InfA_1$ (3,963 Da) in an amount necessary to confer anti-pneumococcal activity but not sufficient to be identified by MALDI-TOF MS. Similarly to $InfA_1$ (3,963Da), $InfA_2$ could have a modified molecular mass which would hinder its identification in this fraction. The use of different purification procedures allowed demonstration of the production of five more of the 13 putative bacteriocins encoded by *S. infantarius* LP90. MS-analyses showed peaks of 4,004 (Fig 5C); 3,709; 4,031; and 5,439 Da (Fig 6A), which approximately matched the predicted molecular sizes of $InfB_1$, $InfC_1$, $InfG_1$, and InfH (Table 3). However, in the case of $InfG_1$ two additional peaks with modified molecular masses (4,067 and 4,051 Da; Fig 5C) were also detected. Considering that $InfG_1$ contains two methionine residues, the peak of 4,067 Da could be the result of the conversion of one Met to MetSO2 (addition of 32 Da) or the conversion of the two Met to methionine sulfoxide (MetSO) (addition of 16 Da each one). Likewise, the lower intensity peak with a molecular mass of 4,051 Da

could also correspond to InfG$_1$ due to the conversion of one Met to MetSO (addition of 16 Da). Moreover, peptides that could correspond to InfB$_2$ were also detected (4,714 Da in Fig 4, and 4,718 Da in Figs 5A and 6A). The molecular mass of these purified peptides is approximately 32 Da higher than that of the predicted one, which might be due to a spontaneous modification of the methionine residue to methionine disulfide (MetSO2) (addition of 32 Da). Considering these results, we can not discard the possibility that other infantaricins be produced but not detected, especially considering that all their structural genes are transcribed. In this respect, some of the bacteriocins produced by *S. infantarius* LP90 could have undergone modifications which would preclude their identification by MALDI-TOF MS.

## Conclusions

*S. infantarius* LP90, a safe dairy strain exerting anti-pneumococcal activity, harbours a 19,539-bp multibacteriocinogenic cluster consisting of 29 ORFs including the genes encoding 8 new class II-bacteriocins (InfA [InfA$_1$-InfA$_2$], InfB [InfB$_1$-InfB$_2$], InfC [InfC$_1$-InfC$_2$], InfD [InfD$_1$-InfD$_2$], InfE, InfF, InfG [InfG$_1$-InfG$_2$] and InfH), as well as the proteins involved in their secretion, immunity and transcriptional regulation. The anti-pneumococcal activity of *S. infantarius* LP90 is due, at least, to a modified InfA$_1$ (3,963Da). In addition to InfA$_1$, biologically active InfB$_1$, InfB$_2$, InfC$_1$, InfG$_1$ and InfH, which lack anti-pneumococcal activity, are also secreted by this strain. InfA$_1$ (3,963Da) displays a broad anti-pneumococcal spectrum and other interesting characteristics and could be employed as an alternative or complementary strategy to antibiotics for the control of human infections caused by *S. pneumoniae*.

## Supporting information

**S1 Fig. Phenotypic detection of virulence factors in *S. infantarius* LP90.** (A) β-hemolysin, (B) gelatinase and (C) protease activity; (1) and (2) *S. infantarius* LP90, (3) *E. faecalis* P4, and (4) *E. faecalis* P36 used as positive controls.
(PPT)

**S2 Fig. Biogenic amine production in *S. infantarius* LP90.** (A) Tyrosine, (B) histidine, and (B) ornithine decarboxylase activities. (1) *S. infantarius* LP90, (2) *Lb. brevis* CECT4121 used as positive control for tyrosine descarboxylase activity, and (3) *Lactobacillus* sp. ATTCC30a used as positive control for histidine and ornithine descarboxylase activities.
(PPT)

**S3 Fig.  Mucine degradation (A) and bile salt deconjugation (B) abilities in *S. infantarius* LP90.** (1) and (2) *S. infantarius* LP90, and (3) fresh fecal slurry from a healthy adult horse as positive control.
(PPT)

**S4 Fig. RP-FPLC chromatography showing the peptides obtained from *a S. infantarius* LP90 CFS subjected to purification procedure P1.** Fractions 26 and 28 eluted at 30 and 32% (v/v) 2-propanol in aqueous TFA (0.1%, v/v), respectively. Red and black dashed lines indicate antimicrobial activity against *S. pneumoniae* 66405 and *P. damnosus* CECT4797, respectively.
(PPT)

**S5 Fig. RP-FPLC chromatography showing the peptides obtained from a *S. infantarius* LP90 CFS subjected to purification procedure P2.** Fractions 15, 17, and 18 eluted at 34, 36, and 38% (v/v) 2-propanol in aqueous TFA (0.1%, v/v), respectively. Red and black dashed lines indicate antimicrobial activity against *S. pneumoniae* 66405 and *P. damnosus* CECT4797,

 

respectively.
(PPT)

**S1 Table. Bacteriocinogenic LAB and indicator microorganisms used in this study.**
(XLS)

**S2 Table. Antimicrobial activity of bacteriocinogenic LAB against *S. pneumoniae* clinical strains.**
(XLS)

**S3 Table. Primers and PCR conditions for detection of virulence factor genes.**
(XLS)

**S4 Table. Primers and PCR conditions for detection of bacteriocin structural genes.**
(XLS)

## Acknowledgments

The authors express their gratitude to Dr. T. J. Eaton and Dr. M. Gasson, Institute of Food Research, Norwich, United Kingdom, for supplying some strains used as PCR positive controls.

## Author Contributions

**Conceptualization:** Dzung B. Diep, Rosa del Campo, Ingolf F. Nes, Pablo E. Hernández, Carmen Herranz, Luis M. Cintas.

**Data curation:** Cristina Campanero, Rosa del Campo.

**Formal analysis:** Ingolf F. Nes, Pablo E. Hernández, Carmen Herranz, Luis M. Cintas.

**Funding acquisition:** Pablo E. Hernández, Luis M. Cintas.

**Investigation:** Cristina Campanero, Estefanía Muñoz-Atienza, Dzung B. Diep, Javier Feito, Sara Arbulu, Rosa del Campo.

**Methodology:** Cristina Campanero, Estefanía Muñoz-Atienza, Dzung B. Diep, Javier Feito, Sara Arbulu, Rosa del Campo.

**Project administration:** Pablo E. Hernández, Luis M. Cintas.

**Resources:** Luis M. Cintas.

**Supervision:** Pablo E. Hernández, Carmen Herranz, Luis M. Cintas.

**Writing – original draft:** Cristina Campanero.

**Writing – review & editing:** Cristina Campanero, Carmen Herranz, Luis M. Cintas.

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
