## [Decision Letter · Decision Letter 0]

10 Dec 2019

PONE-D-19-30374

Biochemical, genetic and transcriptional characterization of multibacteriocin production by the anti-pneumococcal dairy strain Streptococcus infantarius LP90

PLOS ONE

Dear Ass. Prof. Cintas,

Thank you for submitting your manuscript to PLOS ONE. After careful consideration, we feel that it has merit but does not fully meet PLOS ONE’s publication criteria as it currently stands. Therefore, we invite you to submit a revised version of the manuscript that addresses the points raised during the review process.

We noted that both reviewers want more details of experiments and /or suggest to include data which have not been shown in the manuscript pertaining to this study. We expect the authors to follow these suggestions and provide information asked by the reviewers.  Furthermore, make sure the figures are easy to understand (pay attention to comment about Figure 3). This could be achieved by 1) generation high resolution image and 2) by showing the images to other colleagues and see if they can understand and interpret the images without much difficulty. 

We would appreciate receiving your revised manuscript by Jan 24 2020 11:59PM. To enhance the reproducibility of your results, we recommend that if applicable you deposit your laboratory protocols in protocols.io, where a protocol can be assigned its own identifier (DOI) such that it can be cited independently in the future. For instructions see: http://journals.plos.org/plosone/s/submission-guidelines#loc-laboratory-protocols

We look forward to receiving your revised manuscript.

Kind regards,

Anirudh K. Singh, Ph.D

Academic Editor

PLOS ONE

Journal Requirements:

Reviewers' comments:

Reviewer's Responses to Questions

**Comments to the Author**

1. Is the manuscript technically sound, and do the data support the conclusions?

Reviewer #1: Yes

Reviewer #2: Yes

2. Has the statistical analysis been performed appropriately and rigorously? 

Reviewer #1: N/A

Reviewer #2: N/A

3. Have the authors made all data underlying the findings in their manuscript fully available?

Reviewer #1: Yes

Reviewer #2: Yes

4. Is the manuscript presented in an intelligible fashion and written in standard English?

Reviewer #1: Yes

Reviewer #2: Yes

5. Review Comments to the Author

Reviewer #1: The manuscript focus on biochemical, genetic and transcriptional characterization of bacteriocins production by Streptococcus infantarius LP90 strain isolated from Venezuelan water buffalo milk. The work is interesting, and the authors suggest that these bacteriocins may be applicable in human medicine as antimicrobials.

-Line 80: OD600, 600 as subscript.

-Lines 308-309: Could the authors explain which reference gene was used for RT PCR? And which method was applicated for the analysis?

-Lines 331-339 and 342-343: It is necessary to write “not shown” several times? The authors can show positive results.

-Line 440: Could authors show the three peaks obtained?

-Supporting Information S4 Table. S. uberis strains yield Nisin U, Uberolisin and Ubericin A genes and S. uberis strains could be used as control strains.

Reviewer #2: The authors describe the identification of several novel bacteriocins produced by S. infatarius LP90, including a bacteriocin with anti-pneumococcal activity. The authors present data to show that the strain LP90 is safe, and fully characterize the gene cluster encoding several potential bacteriocins. These results support the potential for using bacteriocins as alternatives or supplements to antibiotic treatment for combating human infections. The authors need to address the specific points listed on the attached document.

6. PLOS authors have the option to publish the peer review history of their article (what does this mean?). If published, this will include your full peer review and any attached files.

Reviewer #1: No

Reviewer #2: No

---

## [Author Response · Author response to Decision Letter 0]

13 Jan 2020

PLOS ONE: PONE-D-19-30374

Biochemical, genetic and transcriptional characterization of multibacteriocin production by the anti-pneumococcal dairy strain Streptococcus infantarius LP90

ACADEMIC EDITOR

Dear Dr. Anirudh K. Singh,

Thank you very much for your criticism and constructive comments on our manuscript. As you suggested, a higher resolution Figure 5 has been provided and this image was showed to some colleagues that understood and interpreted it without difficulty. 

Sincerely yours

Ass. Prof. Luis M. Cintas Izarra

REVIEWER #1: 

Dear Reviewer:

Thank you very much for your criticism and constructive comments on our manuscript. Please, find below our point-to-point answers to your comments:

1. Line 80: OD600, 600 as subscript.

Modified as suggested (lines 80, 95 and 300; pages 4, 5 and 13, respectively)

2. Lines 308-309: Could the authors explain which reference gene was used for RT PCR? And which method was applicated for the analysis?

In all the RT-PCR analysis we obtained amplicons of the expected size, which was used as an internal control that the reactions were working properly. On the other hand, with regard to the method applied for the analysis, please note that a paragraph with this information has been included in the revised manuscript (lines 304-310, page 14). 

3. Lines 331-339 and 342-343: It is necessary to write “not shown” several times? The authors can show positive results.

As suggested, these results have been included in the revised manuscript as supplemental files (S1 Fig, S2 Fig and S3 Fig). 

4. Line 440: Could authors show the three peaks obtained?

We are not sure if you want to see only this chromatogram or also the other obtained with the purification procedure 1. Please note that both chromatograms have been included as supplemental files (S4 Fig and S5 Fig).

5. Supporting Information S4 Table. S. uberis strains yield Nisin U, Uberolisin and Ubericin A genes and S. uberis strains could be used as control strains.

We acknowledge that S.uberis strains can be used as bacteriocin positive controls but, unfortunately, we could not have access to these strains.

REVIEWER #2

Dear Reviewer:

Thank you very much for your criticism and constructive comments on our manuscript. Please, find below our point-to-point answers to your comments:

1. Page 7: Table 1, showing susceptibility to antibiotics, is inserted within the Materials and Methods section, but it should be moved to Page 15 where these results are reported in the Results section. A list of the antibiotics used could be provided within the text of the Materials and Methods. 

A list of the antibiotics used has been introduced in the Materials and Methods section (lines 140-143, page 7).

Table 1 has been deleted from “Material and Methods” and introduced in the Results section (line 343, page 15). 

2. Page 15, line 315: It should be mentioned in the text if the 37 LAB screened were novel isolates from this study or a previously reported study. 

The sentence “isolated in previously reported works” has been introduced in the Results section (line 318, page 14).The references are in S1 Table.

Why is reported that LAB inhibited most of the 9 S. pneumoniae strains screened, when 23 clinical isolates were tested (lines 316-318)? 

As stated in the Materials and Methods section, the direct antimicrobial activity of the 37 bacteriocinogenic LAB was screened by a stab-on-agar test (SOAT) against 9 S. pneumoniae strains (lines 83-84, page 4), while their extracellular antimicrobial activity was determined by an agar well-diffusion test (ADT) against 23. S. pneumoniae strains (lines 92-94, page 5).

Was there something unique about these 9 strains, and which 9 strains are being referenced? 

These 9 strains were selected on the basis of their serotypes and antibiotic susceptibility patterns to assure heterogeneity. The information about which 9 strains were used as indicators in SOATs can be found in S2 Table.

Line 327: How many strains were inhibited by LP90, which showed the broadest anti-pneumococcal spectrum? 

Please, note that this information has been included in the revised manuscript (lines 331-332; page 15).

Line 342: It should be made clear that LP90 was screened for the presence of known bacteriocin genes by PCR, and none were identified. 

Please, note that this subject has been clarified in the revised manuscript (lines 346-347; page 16).

3. It should also be said if the streptococcal genes screened for included known bacteriocins from other S. infantarius strains. 

Bacteriocins have not been previously identified in the species S. infantarius. Please, note that this sentence has been included in S4 Table footnote (superscript b explanation).

4. Page 17: Nucleotide sequence of multibacteriocinogenic gene cluster: The labeling of genes are not clear in Figure 3, this image must be generated so the reader can see where the genes and operons are located within the gene cluster. It is difficult to follow the text without the corresponding map of the gene cluster. 

Please note that this figure has been modified to increase its resolution. 

5. Line 432: Pediococcus must be spelled out for strain CECT4797. This is the first time this strain is referenced in the text, but it should include within the Materials and Methods section as well. 

As suggested, Pediococcus has been spelled in the Materials and Methods section (line 262, page 12). 

6. Line 439-444: The fractions reported should be listed specifically with the image shown in Figure 5 (5a, 5b, 5c). The differences between each figure should also be explained with a figure legend. 

Listed as suggested (lines 443-453, page 24). Moreover, the differences between each figure have been explained in Figs 5 legends. Please, note that the predominant peaks are pointed with an arrow in the modified Figs 5.

Two lower intensity peaks are reported for fractions B and C, but several other molecular weights are shown, why were these two peaks reported? 

This sentence has been deleted. Because the reported peaks have molecular masses which could be corresponded to modified (oxidized) bacteriocins as indicated in the Discussion section (lines 521-530, page 27 and 28).

7. Line 447: Within the text the different fractions shown in Figure 6 (a and b) should be specified. 

The different fractions shown in Figs 6a and b have been specified as suggested (lines 459-460, page 25). 

Please, note that the letter “a” has been added to Fig.6 (line 526, page 27; and line 536, page 28).

8. Line 477: Is it correct to say the potential bacteriocins “would share the same secretion and regulation system”? Is this based on the presence of similar signal peptides on bacteriocins, or just on the fact that they are co-located within the same gene cluster? The authors need to be clear when making conclusions within the Discussion. 

The bacteriocins are clustered and flanked by secretion and regulation genes, which supports that they would share the same secretion and regulation systems.

Line 486-487: Which bacteriocins are potentially class IIa and IIe, since three bacteriocins are listed.

This has been clarified in the revised manuscript (lines 496-497, page 26).

9. Line 496-501: What is shown that orf29 encoded an IF, or was this conclusion based on sequence similarity? 

This statement is based on sequence similarity. Please, note that this has been clarified in the revised manuscript (line 506-507, page 27).

Are the molecular masses variants of InfA1 or could they be separate peptides? This section, describing the peptides with molecular masses of 3, 963; 3,354 and 3.357 Da, needs to be discussed more clearly.

We agree that these two peptides could be separate peptides and have mentioned this possibility in the revised manuscript (lines 508-509, page 27). However, we do consider more likely the hypothesis we present in the manuscript. With regard to this, we would like to note that some bacteriocins require a non-peptidic moiety to exert antimicrobial activity. On the other hand we have done our best to modify the section describing the peptides with molecular masses of 3, 963 and 3,354 Da in order to make it more clear for the reader (lines 508-522, page 27).

---

## [Decision Letter · Decision Letter 1]

6 Feb 2020

Biochemical, genetic and transcriptional characterization of multibacteriocin production by the anti-pneumococcal dairy strain Streptococcus infantarius LP90

PONE-D-19-30374R1

Dear Dr. Cintas,

We are pleased to inform you that your manuscript has been judged scientifically suitable for publication and will be formally accepted for publication once it complies with all outstanding technical requirements.

With kind regards,

Anirudh K. Singh, Ph.D

Academic Editor

PLOS ONE

Additional Editor Comments (optional):

Reviewers' comments:

Reviewer's Responses to Questions

**Comments to the Author**

1. If the authors have adequately addressed your comments raised in a previous round of review and you feel that this manuscript is now acceptable for publication, you may indicate that here to bypass the “Comments to the Author” section, enter your conflict of interest statement in the “Confidential to Editor” section, and submit your "Accept" recommendation.

Reviewer #1: All comments have been addressed

2. Is the manuscript technically sound, and do the data support the conclusions?

Reviewer #1: Yes

3. Has the statistical analysis been performed appropriately and rigorously? 

Reviewer #1: Yes

4. Have the authors made all data underlying the findings in their manuscript fully available?

Reviewer #1: Yes

5. Is the manuscript presented in an intelligible fashion and written in standard English?

Reviewer #1: Yes

6. Review Comments to the Author

Reviewer #1: The manuscript describes a novel anti-pneumococcal bacteriocin. All comments have been addressed to the manuscript and it is now ready for publication.

7. PLOS authors have the option to publish the peer review history of their article (what does this mean?). If published, this will include your full peer review and any attached files.

Reviewer #1: No

---

## [Editor Report · Acceptance letter]

10 Feb 2020

PONE-D-19-30374R1 

Biochemical, genetic and transcriptional characterization of multibacteriocin production by the anti-pneumococcal dairy strain *Streptococcus infantarius* LP90 

Dear Dr. Cintas:

I am pleased to inform you that your manuscript has been deemed suitable for publication in PLOS ONE. Congratulations! Your manuscript is now with our production department. 

With kind regards,

on behalf of

Dr. Anirudh K. Singh 

Academic Editor

PLOS ONE